# Cardiovascular Comorbidities in Chronic Obstructive Pulmonary Disease (COPD)—Current Considerations for Clinical Practice

**DOI:** 10.3390/jcm8010069

**Published:** 2019-01-10

**Authors:** Frederik Trinkmann, Joachim Saur, Martin Borggrefe, Ibrahim Akin

**Affiliations:** 11st Department of Medicine (Cardiology, Angiology, Pulmonary and Intensive Care), University Medical Center Mannheim, Medical Faculty Mannheim, Heidelberg University, 68167 Mannheim, Germany; jsaur@uni-mannheim.de (J.S.); martin.borggrefe@umm.de (M.B.); ibrahim.akin@umm.de (I.A.); 2DZHK (German Center for Cardiovascular Research), partner site Mannheim, University Medical Center Mannheim, Medical Faculty Mannheim, Heidelberg University, 68167 Mannheim, Germany

**Keywords:** COPD, comorbidities, cardiovascular, diagnostics, therapy

## Abstract

In patients with chronic obstructive pulmonary disease (COPD), cardiovascular comorbidities are highly prevalent and associated with considerable morbidity and mortality. This coincidence is increasingly seen in context of a “cardiopulmonary continuum” rather than being simply attributed to shared risk factors such as cigarette smoking. Overlapping symptoms such as dyspnea or chest pain lead to a worse prognosis due to missed concomitant diagnoses. Moreover, medication is often withheld as a result of unfounded concerns about side effects. Despite the frequent coincidence, current guidelines are still mostly restricted to the management of the individual disease. Future diagnostic and therapeutic strategies should therefore be guided by an integrative perspective as well as a refined phenotyping of disease entities.

## 1. Introduction

Concomitant cardiovascular disease in chronic obstructive pulmonary disease (COPD) is increasingly seen in context of a “cardiopulmonary continuum” [1] rather than being simply attributed to shared risk factors such as cigarette smoking. In recent years, complex cardio-respiratory interactions were identified. These are not restricted to structural, vascular, and genetic factors. Both disease entities are centrally linked to systemic inflammation (Figure 1). Various factors contribute to this process, most notably inhaled noxae but also hypoxia, oxidative stress, ageing, and reduced physical activity. Inflammatory levels are further increased by disorders which themselves lead to aggravation or development of comorbidities. These include atherosclerosis and obstructive bronchitis, being the basis for acute events such as myocardial infarction or acute exacerbations. Structural changes during remodeling eventually lead to organ failure which clinically presents as heart failure or respiratory failure, respectively.

Different clinical phenotypes of COPD may be associated with specific inflammatory signaling pathways. While cardio-metabolic disease frequently leads to airway-predominant COPD, sarcopenia, and osteoporosis is predominantly found in patients with emphysema. In four out of five COPD patients, at least one comorbidity is present, with cardiovascular entities being most frequent [2,3]. This multimorbidity is a major challenge to health care systems world-wide, requiring an integrative rather than highly specialized approach.

## 2. Cardiovascular Risk

Patients with COPD have a two- to three-fold increased cardiovascular morbidity and mortality [5]. It is associated with disease severity [6], and inflammation is seen to be particularly important. This finds its expression in elevated markers of inflammation even in stable pulmonary disease [7], indicating a link between local pulmonary and systemic inflammatory processes. Hence, therapeutic interventions based on inhaled corticosteroids (ICS) modulating systemic inflammation seem attractive. Being a cornerstone of anti-inflammatory therapy in bronchial asthma, their use is restricted to distinct phenotypes in COPD. Main indications include patients with frequent exacerbations or blood eosinophilia according to current Global Initiative for Chronic Obstructive Lung Disease (GOLD) recommendations [8]. Presence of eosinophilic inflammation can be assessed using fractional exhaled nitric oxide (FeNO). Recently, it was shown that patients with high FeNO levels and non-specific respiratory symptoms were more likely to respond to ICS as compared to placebo [9]. However, no differences in cardiovascular events were found between patients at risk treated with either fluticasone furoate, vilanterol, or their combination as compared to placebo [10]. Although therapy did not alter plasma cardiac troponin I levels, the latter were recently shown to predict future cardiovascular events [11]. In contrast, studies targeting cardiovascular safety of oral phosphodiesterase inhibitor roflumilast revealed a 35% reduction of cardiovascular events [12]. Likewise, a decrease in hyperinflation was associated with a reduction of systemic inflammation. This may explain the positive effect of dual bronchodilation as compared to long-acting beta agonist (LABA)/ICS combination therapy that are still used frequently [13].

Statins are primarily used for reduction of serum cholesterol levels. Apart from this, pleiotropic immune modulatory as well as anti-inflammatory effects have been described [14,15]. These may independently contribute to a decrease of airway inflammation in theory. A 38% reduction of overall mortality and 31% reduction of myocardial infarction was previously found in a systematic review of observational research [16]. Although there seems to be a correlation between reduction of forced expiratory volume in one second (FEV_1_) and cardiovascular mortality [17], statin treatment is neither associated with improved pulmonary nor vascular function in general [18]. Evidence from observational studies towards a lower exacerbation rate could not be consistently confirmed in controlled trials [19]. However, there was a positive effect in a prespecified subgroup of patients with supra-median inflammatory levels [18]. A decrease in systemic inflammation markers was seen in 79% of patients receiving pravastatin in a randomized trial and associated with improvement in exercise time as compared to placebo [20]. Statin doses have been a major point of criticism. Patients with high inflammatory levels particularly seem to benefit most from therapy [21]. Although being potentially attractive, these contradicting study results suggest that the concept does not completely capture the complex disease interactions. Therefore, improved phenotyping of COPD may be a starting point for innovative protocols of urgently needed controlled trials. 

Problematically, current risk stratification scores do not sufficiently include the increased cardiovascular risk associated with COPD [22]. In contrast to primary prevention, secondary prevention statin therapy is associated with a decrease in cardiovascular events and mortality [23]. However, this patient group is neither addressed by the target value approaches recommended in the 2016 European Society of Cardiology (ESC)/European Atherosclerosis Society (EAS) [24] and 2017 American Association of Clinical Endocrinologists (AACE)/American College of Endocrinology (ACE) guidelines [25] nor by the statin benefit group approach that is still currently pursued by 2013 American College of Cardiology (ACC)/American Heart Association (AHA) guideline [26]. COPD itself is increasingly recognized as an independent cardiovascular risk factor, in addition to coronary heart disease, peripheral arterial occlusive disease, diabetes mellitus, or renal failure [27,28]. Hence, statin therapy may be justified in patients with COPD for secondary prevention [29]. However, it should be noted that randomized controlled trials are lacking. Modification of the cardiovascular risk profile outside current guideline recommendations is a challenging task. It requires individual and informed consent between patients and physicians.

## 3. Coronary Heart Disease

Endothelial dysfunction fundamentally contributes to the development of atherosclerosis that finally leads to coronary heart disease (CHD). The process is further accelerated by systemic inflammation and oxidative stress. As a result, approximately one out of six COPD patients suffers from concomitant CHD [30,31]. Moreover, COPD exacerbations are associated with a transient deterioration of endothelial function. This translates into an increased risk for macrovascular complications such as myocardial infarction and stroke. Additionally, subsequent loss of lung function is associated with a long-term increase in arterial stiffness [32]. In the acute setting, all-cause mortality can be predicted by elevated levels of cardiac troponin [33]. Coronary artery calcification correlates with dyspnea, exercise capacity and all-cause mortality. This indicates a link between coronary heart disease and poor clinical outcome in patients with COPD [34]. Therefore, identification of patients with a high coronary artery calcium score is important in order to provide appropriate targeted therapy and cardiovascular risk modification. Current guidelines are mostly restricted to the individual cardiac or respiratory disease [8,35]. Nevertheless, an integrative perspective is warranted, especially as long-term data of patients with COPD and CHD is scarce. Smoking cessation still is the most important measure for secondary prevention and will remain so in the foreseeable future. Concomitant COPD is not diagnosed in around 80% of patients undergoing coronary intervention [36]. This mostly affects early or moderate stages in which better preventive and therapeutic options would still be available. Vice versa, electrocardiographic signs of previous myocardial infraction are not recognized in 70% of patients presenting with acute COPD exacerbations [37]. Clinical assessment is impeded by frequently present atypical angina pectoris, dyspnea or palpitations leading to misinterpretation [4].

Beta blockers are a corner stone of drug treatment in stable CHD while being associated with symptom control and improved prognosis. Nevertheless, they are often withheld or under-dosed in COPD patients. This is a result of unfounded concerns about side effects such as aggravation of dyspnea or bronchoconstriction [38,39] and associated with worse prognosis after myocardial infarction. Cardio-selective beta blockers were shown to be safe when (partly)-reversible obstruction is present [40]. Retrospective analyses even suggest a reduced mortality and exacerbation frequency in this setting [41]. Ivabradine is a reasonable substitute in patients with sinus rhythm intolerant to beta blockers also improving diagnosis. Calcium channel blockers and nitrates are alternatives in patients with angina pectoris [42]. Medication inhibiting the renin–angiotensin–aldosterone system (RAAS) are indicated in patients with heart failure, arterial hypertension or diabetes mellitus and reduce morbidity as well as mortality [35]. Additionally, positive effects were postulated on systemic inflammation, skeletal muscle function, peripheral oxygen consumption, and erythrocytopoiesis [43]. Although these could not be fully confirmed in interventional trials [44], potential benefits should be interpreted in the context of reducing cardiovascular mortality and warrant further investigation. Moreover, no evidence for an increased incidence of bronchoconstriction or cough exists [45,46]. While acetylsalicylic acid is primarily associated with an increased exacerbation risk in asthma, reversible P2Y₁₂-antagonists (ticagrelor, cangrelor) are suspected to cause dyspnea. In contrast, irreversible substances such as clopidogrel and prasugrel do not seem to be associated with increased respiratory symptoms [47].

Still, there is no evidence that would justify deviation from current therapeutic recommendations in patients with concomitant COPD and CHD [8]. Concerns about the cardiovascular safety of beta mimetic or antimuscarinic drugs should always be evaluated in context of an inherently increased cardiovascular risk in COPD. During initiation of LABAs and anticholinergics, an increased risk of cardiovascular events was found in a retrospective cohort study in the elderly [48]. Moreover, betaagonists may precipitate ischemia, congestive heart failure, arrhythmias, and sudden death [49]. Apart from the beforementioned SUMMIT study, meta-analyses even suggest a reduction of severe cardiovascular events by tiotropium bromide [50]. Likewise, it could be demonstrated that long-acting muscarinic antagonists were associated with a reduced risk of acute myocardial infarction when applied via a dry powder inhaler. Beta agonists alone were associated with an increased risk of acute myocardial infarction while combination with ICS was not [51]. In general, long-acting substances have an acceptable safety profile [52]. However, it seems reasonable to avoid high doses of short-acting drugs during acute coronary syndrome (Table 1).

Myocardial revascularization is associated with worse long-term results in patients with concomitant COPD [53,54]. This may be the clinical expression of rather diffuse lesions [55]. These are harder to tackle using both surgery (coronary arterial bypass graft, CABG) as well as interventional (percutaneous coronary intervention, PCI) approaches. Moreover, COPD patients are less likely to receive guideline-based medical therapy. This may further contribute to the increased mortality as well as revascularization rates. When choosing between CABG vs. PCI as the optimal approach, perioperative risk (sternotomy, possibly extracorporeal circulation) has to be carefully evaluated in context of higher rates of restenosis and the lack of data showing an improved prognosis. Risk–benefit assessment should be performed in interdisciplinary Heart Teams guided by prediction scores such as SYNTAX II or EuroSCORE II. Both include COPD as independent risk factors [56].

## 4. Heart Failure

Left heart failure is diagnosed first-time in one out of five COPD patients after extensive cardiovascular work-up [64]. Vice versa, one out of three heart failure patients suffers from obstructive ventilation disorders [65]. These frequent coincidences are not only associated with diagnostic difficulties, but also with worse prognosis (five-year survival 31% vs. 71%) [66]. Concomitant obstructive ventilation disorders during acute left heart failure are fully reversible in half of the cases after six months. However, initial hyperinflation has a predictive value for diagnosis of COPD [67]. Elevated brain natriuretic peptide concentrations (BNP) above 500 pg/mL are more likely to be caused by left heart failure in acute settings [68]. In contrast, pulmonary differential diagnoses are more frequently found if levels are below 100 pg/mL. Between these thresholds, diagnostic performance for detection of biventricular heart failure and right heart failure alone is limited. In general, the latter tends to lower BNP values [69]. Evaluation of patient history as well as clinical signs can be valuable in differential diagnosis. For example, acute exacerbations of COPD often lead to apical and ventral wheezing. In contrast, alveolar edema due to left heart decompensation leads to fine crackles but also wheezing of lower parts of the lung. Moreover, interpretation of conventional X-ray is complicated by chronic structural changes. At the same time, they can superimpose or mimic pulmonary venous congestion. Hence, evaluation of pulmonary and cardiac limitations should only be done in stable, euvolemic phases of about three months’ time frame. Although no robust data is available evaluating the efficacy of screening programs, both organ systems seem worth to be included in diagnostic considerations. This holds especially as non-invasive diagnostic tools as well as treatment strategies are widely available. In stable COPD, transthoracic echocardiography should be initially performed. However, poor acoustic window may be present in up to 50% the patients depending on the degree of airflow limitation [70]. Evaluation of heart failure with preserved ejection fraction (HFpEF) is particularly challenging in this setting and associated with false diagnoses due to pulmonary comorbidity [71]. Alternatives include cardiac magnetic resonance imaging as well as cardio pulmonary exercise testing. The latter additionally allows functional phenotyping of the disease [72]. Natriuretic peptides are primarily used for excluding heart failure at thresholds below 125 pg/mL for NT-pro-BNP and 35 pg/mL for BNP, respectively [73]. Elevated values should be further evaluated.

There is currently no evidence to pursue treatment strategies differing from current guidelines for COPD with concomitant heart failure. However, beta blockers are still frequently withheld in this setting although they were shown to considerably reduce mortality [74]. Cardio selective drugs should be preferred if possible [75]. As compared to unselective substances such as carvedilol, bisoprolol was shown to cause fewer side effects and even improve lung function [76]. The value of novel therapeutic options such as angiotensin receptor–neprilysin inhibition (sacubitril/valsartan) in patients with heart failure and concomitant COPD warrants further evaluation. Recently, dual bronchodilator therapy was demonstrated to improve cardiac function. It may therefore warrant early medical treatment in COPD patients with signs of hyperinflation [77]. Table 1 summarizes indications and possible pitfalls of frequently used medication in patients with COPD and cardiovascular disease, respectively.

## 5. Arterial Hypertension

Arterial hypertension is comparably frequent in COPD patients and the general population. Prevalence is about 50% and increasing with age [78]. Nevertheless, the overall elevated cardiovascular risk in COPD may be a product of this common condition and potentiating effects of other risk factors such as diabetes mellitus and cigarette smoking. Moreover, higher central blood pressure values and arterial stiffness are also found indicating premature atherosclerosis [79]. Central blood pressure values are more strongly correlated with markers of hypertensive end-organ damage in general [80,81,82] and better predict cardiovascular outcomes [81,83]. Moreover, important differences between classes of antihypertensive drugs were found regarding their effect on central blood pressure in general populations [84,85], despite having a similar impact on brachial values [86,87]. However, there is currently no evidence for application of alternative blood pressure target values or medication when arterial hypertension is present in COPD patients. Theoretical advantages of calcium antagonists due to smooth muscle relaxation only translate into little clinical effect [88]. In recent years, several techniques for non-invasive determination of central blood pressure became available [89,90,91]. They have the potential to facilitate application in clinical routine. However, it remains to be determined whether measurement of central blood pressure will also improve differential antihypertensive therapy in patients with COPD.

## 6. Pulmonary Hypertension

Increased pulmonary blood pressure can be most commonly attributed to left heart disease (Nizza group 2) or pulmonary disease (Nizza group 3). In the latter, development of pulmonary hypertension is associated with a worse prognosis than in patients with COPD alone. Interestingly, there is a morphological overlap between lesions found in group 1 (pulmonary arterial hypertension, PAH) and group 3 pulmonary hypertension, respectively. The five-year survival rate (36%) is primarily driven by hemodynamic and not ventilatory parameters [92]. Likewise, a two-fold increase in mortality is found in patients with left heart disease (group 2). This is due to consecutive right heart failure and to date convincing therapeutic concepts are missing. When mitral valve disease is present additionally, surgical and interventional options should be evaluated as they may lead to considerable reduction in pulmonary arterial pressure [93]. In both groups 2 and 3, therapeutic response to targeted therapy as used in PAH (group 1) seems to be closely related to hemodynamic impairment. Identification of these subgroups requires a profound diagnostic work-up and should ideally be performed in expert centers following urgently required study protocols [92].

## 7. Cardiac Arrhythmias

Atrial fibrillation is common during acute COPD exacerbations and may complicate differential diagnosis [94]. Overall, there is an 1.8-fold increase in prevalence in COPD patients that is associated with the degree of lung function impairment [95]. Elevated pulmonary arterial pressures with consecutive right heart failure are made responsible. This is a result of hypoxemia, hypercapnia, and systemic inflammation, but also drug side effects. When initiating oral steroids or using high doses (≥7.5 mg prednisolone or equivalent), a 3.4-fold increase in risk for newly onset atrial fibrillation was seen [58]. In contrast, ICS was not associated with a higher risk [57]. In contrast to short-time use of theophylline, beta agonists were not associated with atrial fibrillation in this large case–control study (Table 1). Compensation of hypoxemia and respiratory acidosis are cornerstones of therapy. Both conditions were shown to negatively influence the effectivity of medication as well as electrical cardioversion [96]. Vice versa, the presence of COPD itself is a risk factor for unsuccessful catheter ablation therapy [97] as well as for recurrent atrial fibrillation after electrical cardioversion [98]. This may be due to anatomical changes in pulmonary circulation which are associated with right atrial foci in COPD patients [99]. Being less susceptible to electrical cardioversion, secondary multifocal atrial tachycardias have to be clearly differentiated from atrial fibrillation [100]. Apart from optimizing COPD therapy, verapamil or beta blockers are specific therapeutic options. Given the tremendous arrhythmogenic potential and availability of effective inhaled drugs, it should be possible to avoid theophylline effortlessly. However, caution may be warranted in short-acting bronchodilators. In contrast, benefits most often exceed the potential risks of long-acting substances. Non-dihydropyridine calcium channel blockers and cardio-selective beta blockers are considered to be safe for frequency control in COPD patients [40]. In contrast, class IA and IC antiarrhythmic drugs are often contraindicated due to concomitant CHD.

The overall elevated risk for sudden cardiac death further increases after five and a half years from initial diagnosis as well as in patients with frequent exacerbations [101]. Most often, ventricular arrhythmias are made responsible. Likewise, short-acting beta agonists, theophylline and oral steroids have arrhythmogenic potential due to a decrease in potassium concentrations [49,57]. Previous findings suggest that patients with COPD more frequently die during night time. This is potentially due to the reduced ventilation resulting in more ventricular ectopic episodes while they are asleep. Hence, therapeutic options in current guidelines include medication (beta blockers, amiodarone), implantable cardioverter defibrillator (ICD), and catheter ablation, but also cautious handling of QT-prolongating drugs [102]. Pulmonary toxicity, especially development of pulmonary fibrosis should be taken into consideration for long term amiodarone treatment. Incidence is dose dependent and estimated as high as 15%. Typical daily doses below 200 mg are associated with a 0.1 to 0.5% risk [63]. Therefore, symptoms as well as transfer factor should be monitored routinely. Sotalol is a class III antiarrhythmic alternative. However, it has the potential to cause bronchoconstriction due to its non-selective binding to both β1- and β2-adrenergic receptors.

## 8. Summary

Concomitant cardiovascular disease is frequent in COPD patients and associated with considerable morbidity and mortality. In recent years, complex cardio-respiratory interactions were identified. Rather than being simply attributed to shared risk factors such as cigarette smoking, coincident cardiovascular disease and COPD is increasingly seen in context of a “cardiopulmonary continuum”. Both entities are centrally linked to systemic inflammation. Current guidelines are still mostly restricted to the management of the individual disease. However, overlapping symptoms such as dyspnea or chest pain lead to a worse prognosis due to missed concomitant cardiac or pulmonary diagnoses. Their presence should hence lead to further evaluation using lung function testing, echocardiography and electrocardiography. Moreover, medication is often withheld due to unfounded concerns about side effects. Cardio-selective betablockers and long-acting bronchodilators both show an acceptable safety profile. In contrast, theophylline as well as high doses of short-acting bronchodilators and oral corticosteroids contain relevant proarrhythmic potential. Although drug therapy using steroids or statins seems to be attractive for influencing systemic inflammation, the concept does not fully capture the complex interactions. Various additional factors such as inhaled noxae, hypoxia, oxidative stress, ageing, and reduced physical activity contribute to this process. Different clinical phenotypes of COPD may be associated with specific inflammatory signaling pathways. Cardio-metabolic disease frequently leads to airway-predominant COPD. In contrast, sarcopenia and osteoporosis is predominantly found in patients with emphysema. Therefore, future diagnostic and therapeutic strategies should be guided by an integrative perspective as well as a refined phenotyping of the disease entities.

## Figures and Tables

**Figure 1 jcm-08-00069-f001:**
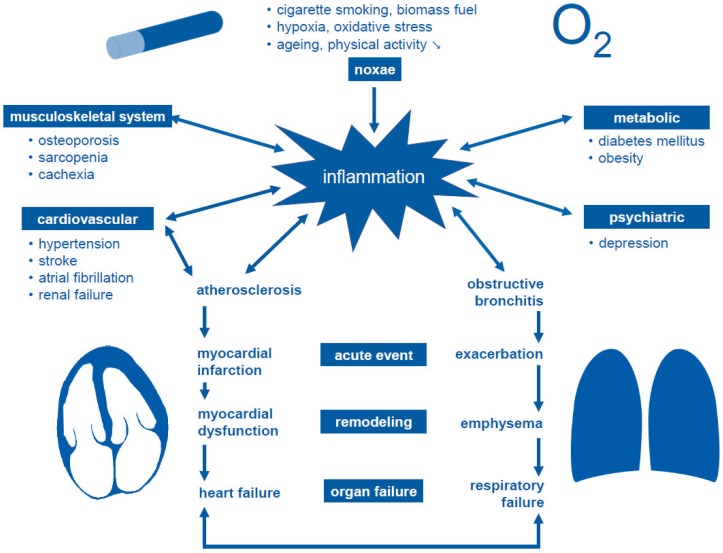
Cardiovascular and pulmonary disease in the context of inflammation (“cardiopulmonary continuum”, modified after [1,4]).

**Table 1 jcm-08-00069-t001:** Frequently used medication in COPD and cardiovascular disease.

	Medication		Indication	Comment	References
**COPD**	steroids	inhaled	long-term therapy ^#^	good safety profile	[57]
		systemic	exacerbation	pro-arrhythmic potential	[58]
	beta agonists	short acting	exacerbation long-term therapy	pro-arrhythmic potential (high doses)	[50,51,52,59]
		long acting	long-term therapy	acceptable safety profile	
	muscarinic antagonists	short acting	exacerbation long-term therapy	pro-arrhythmic potential (high doses)	[50,52,60]
		long acting	long-term therapy	acceptable safety profile	
	PDE inhibitors	Roflumilast	long-term therapy	reduction of cardiovascular events	[12]
		Theophylline	(long-term therapy)	narrow therapeutic range and considerable pro-arrhythmic potential	[57,61]
**Cardiovascular Disease**	beta blockers	selective	heart failure, CHD, ACS, AHT, SVT, VT	often withheld or under dosed, prefer selective substances, overall good safety profile	[40,41]
		non-selective			
	antiplatelet therapy		CHD, ACS	dyspnea caused by reversible P2Y₁₂-antagonists (Ticagrelor, Cangrelor)	[47]
	Ivabradine		CHD, heart failure	alternative to beta blockers for anti-anginal and frequency control (sinus rhythm only)	[42,62]
	statins		CHD, dyslipidemia	secondary prevention, pleiotropic effects of immune system and inflammation	[14,15]
	ACE inhibitors		AHT, heart failure	no bronchoconstriction	[45,46]
	angiotensin receptor blockers			alternative (ACE inhibitor induced cough)	
	calcium channel blockers		AHT, CHD, SVT	alternative to beta blockers for anti-anginal and frequency control, smooth muscle relaxation (small clinical effect)	[42]
	nitrates		CHD	alternative to beta blockers for anti-anginal control	[42]
	amiodarone		SVT, VT	pulmonary toxicity	[63]

COPD: chronic obstructive pulmonary disease, PDE: phosphodiesterase, ACS: acute coronary syndrome, AHT: arterial hypertension, SVT: supraventricular tachyarrhythmia, VT: ventricular supraventricular, CHD: coronary heart disease, ACE: angiotensin converting enzyme. ^#^ in selected patients with frequent exacerbations or blood eosinophilia (see text for details).

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
