# Peer review of "Cardiovascular Comorbidities in Chronic Obstructive Pulmonary Disease (COPD)—Current Considerations for Clinical Practice"

_jcm, 2019, doi:10.3390/jcm8010069_

Reviewer 1 Report

The manuscript has been much improved and is in a nice condition now.

Reviewer 2 Report

The authors answered to my questions arised from the first version of this paper; I think that the paper is now substantially ameliorated and It is suitable to be published.I am  honored to act as a reviewer for JCM. I whish you a merry Cristhmas and a happy new year.

This manuscript is a resubmission of an earlier submission. The following is a list of the peer review reports and author responses from that submission.

Round  1

Reviewer 1 Report

Summary: 

The authors review cardiovascular comorbidities in COPD from the perspective of clinical practice. The cardiovascular comorbidities discussed here include the coronary heart disease, heart failure, arterial hypertension, pulmonary hypertension, and the arrhythmias. The authors cover the prevalence, risk factors, pathophysiological mechanisms, underdiagnosis facts, and medication treatment of the most common cardiovascular comorbidities. 

 Major concerns: 

English language and style in this manuscript needs a lot of work. Some paragraph and sentences are confusing. 

e.g. Line 91, “Vice versa, electrocardiographic signs of previous myocardial infraction are overseen in 70% of patients presenting with acute COPD exacerbations [33]”. From the reference 33, it looks the authors are trying to discuss the underdiagnosis of coronary heart disease in COPD. The word “Overseen” is misused here.

Line 27: Figure 1 shows the mechanism why cardiovascular diseases are associated closely to COPD. It is good. But there is only one simple sentence in the “Introduction” section about the figure. I think the authors could provide more details on the mechanisms how the systemic inflammation is linked to both cardiovascular diseases and COPD.
Line 44: Please clarify in this sentence that in the SUMMIT trial, there is no difference between the placebo, fluticasone furoate, vilanterol and their combination.
Line 56-57: “statin treatment is neither associated with improved pulmonary nor vascular function [15]”.  The [15] referes to a randomized controlled study of rosuvastatin reporting that “in a prespecified subgroup analysis of patients with a supra-median circulating hsCRP concentration (>1.7 mg L-1), rosuvastatin was associated with improved endothelium- dependent vascular function (13% vs. 2%, P = 0.026).” I think this information should also be provided to the readers.
And Pravastatin was reported to increase exercise time of COPD patients in another randomized, double-blinded study (Lee, Tsung-Ming, Mei-Shu Lin, and Nen-Chung Chang. "Usefulness of C-reactive protein and interleukin-6 as predictors of outcomes in patients with chronic obstructive pulmonary disease receiving pravastatin." The American journal of cardiology 101, no. 4 (2008): 530-535.).
Line 82-84: This sentence is very confusing. Please clarify the relationship between troponin, loss of lung function, acute exacerbation and arterial stiffness. The [29] in this sentence refers to a paper about troponin in acute exacerbation of COPD, while the [28] in the same sentence refers to a study about arterial stiffness and emphysema severity in COPD. These two references talked about two different topics of COPD. Is there any reason to put them together in the same sentence?
Line 98-100: The [36] refers to a meta-analysis with the conclusion” beta2-Agonist use in patients with obstructive airway disease increases the risk for adverse cardiovascular events. …  beta2-agonists may precipitate ischemia, congestive heart failure, arrhythmias, and sudden death.” The paper also stated that “In contrast, cardioselective beta-blocker therapy is safe in patients with obstructive lung disease and is associated with significant reductions in cardiovascular mortality.”

Author Response

English language and style in this manuscript needs a lot of work. Some paragraph and sentences are confusing. e.g. Line 91, “Vice versa, electrocardiographic signs of previous myocardial infraction are overseen in 70% of patients presenting with acute COPD exacerbations [33]”. From the reference 33, it looks the authors are trying to discuss the underdiagnosis of coronary heart disease in COPD. The word “Overseen” is misused here.

Language and style were revised intensively throughout the whole manuscript as requested.

Line 27: Figure 1 shows the mechanism why cardiovascular diseases are associated closely to COPD. It is good. But there is only one simple sentence in the “Introduction” section about the figure. I think the authors could provide more details on the mechanisms how the systemic inflammation is linked to both cardiovascular diseases and COPD.

We would like to thank the reviewer for the positive comment on Figure 1 in general. As proposed, we

added a more detailed description if the link between systemic inflammation and cardiovascular disease

as well as COPD to the “Introduction” section.

Line 44: Please clarify in this sentence that in the SUMMIT trial, there is no difference between the placebo, fluticasone furoate, vilanterol and their combination.

We added a statement regarding placebo control to the sentence.

Line 56‐57: “statin treatment is neither associated with improved pulmonary nor vascular function [15]”. The [15] refers to a randomized controlled study of rosuvastatin reporting that “in a prespecified subgroup analysis of patients with a supra‐median circulating hsCRP concentration (>1.7 mg L‐1), rosuvastatin was associated with improved endothelium‐ dependent vascular function (13% vs. 2%, P = 0.026).” I think this information should also be provided to the readers. And Pravastatin was reported to increase exercise time of COPD patients in another randomized, double‐blinded study (Lee, Tsung‐Ming, Mei‐Shu Lin, and Nen‐Chung Chang. "Usefulness of C‐reactive protein and interleukin‐6 as predictors of outcomes in patients with chronic obstructive pulmonary disease receiving pravastatin." The American journal of cardiology 101, no. 4 (2008): 530‐535.).

We revised the paragraph accordingly adding the important statement that vascular function was

improved in the prespecified subgroup with supra‐median inflammatory levels (Neukamm et al).

Moreover, the reference by Lee et al. was added and discussed.

Line 82‐84: This sentence is very confusing. Please clarify the relationship between troponin, loss of lung function, acute exacerbation and arterial stiffness. The [29] in this sentence refers to a paper about troponin in acute exacerbation of COPD, while the [28] in the same sentence refers to a study about arterial stiffness and emphysema severity in COPD. These two references talked about two different topics of COPD. Is there any reason to put them together in the same sentence?

We agree that this sentence is confusing and adapted the paragraph accordingly for clarification.

Line 98‐100: The [36] refers to a meta‐analysis with the conclusion” beta2‐Agonist use in patients with obstructive airway disease increases the risk for adverse cardiovascular events. … beta2‐agonists may precipitate ischemia, congestive heart failure, arrhythmias, and sudden death.” The paper also stated that “In contrast, cardioselective beta‐blocker therapy is safe in patients with obstructive lung disease and is associated with significant reductions in cardiovascular mortality.”

We agree that there is ongoing dispute about the safety of inhaled bronchodilators. While most data is available from retrospective analysis, the increased cardiovascular risk found in theses studies has always to be evaluated in context of an inherently increased cardiovascular risk in COPD itself. We added the conclusions concerning beta2‐agonists in the same section below (lines 145‐153 in the revised version). Moreover, discussion especially during initiation in the elderly was added showing an increased cardiovascular risk. In contrast, long‐acting muscarinic antagonists are not associated with an increased risk. However, it should be noted that high‐quality prospective studies are still lacking. The summary of our paragraph was slightly adapted now recognizing the differential therapeutic aspects. All relevant references were also added to Table 1 (please also see our comment to We thoroughly revised Table 1 and added references to every comment as proposed. Moreover, a statement concerning the long‐term use of ICS was added (please also see our answer to Reviewer 2, point 3 below).

Reviewer 2 Report

A reference to an article in Press does not seem to be adequate documentation.
I have some minor suggestions for improvement.
1. The latest products should be cited in this connection here.
These two references talked about latest topics of COPD.
Ref.5  →    https://thorax.bmj.com/content/69/8/718.short

Ref.12 → https://www.ahajournals.org/doi/abs/10.1161/circresaha.116.308537

2. Figure 1, What meaning did the arrows refer to ?
Please clarify the double-headed arrow in figure 1.
Author should describe more details explain of inflammation mediated COPD mechanism.
3. Lane 132-133
This table 1 is very indistinct. Please include the reference number in the table.
4.Lane 237-248
This paper lacks in-depth summary.
Please discuss that why cardiovascular diseases are associated closely to COPD
and inflammation.  I recommend to revise the manuscript so that the purpose appears
more clearly.
I will appreciate it if you accept all suggested corrections, and resubmit your revised manuscript.

Author Response

A reference to an article in Press does not seem to be adequate documentation. 1. The latest products should be cited in this connection here. These two references talked about latest topics of COPD.

Ref.5 → https://thorax.bmj.com/content/69/8/718.short

Ref.12 → https://www.ahajournals.org/doi/abs/10.1161/circresaha.116.308537

Both references were added to sections “2. Cardiovascular risk” and “3. Coronary heart disease”, respectively. Moreover, were thoroughly updated the reference list. The 2018 update of GOLD report still is only available as an online resource. Apart from this, we did not find any unpublished / in press references any more.

2. Figure 1, What meaning did the arrows refer to? Please clarify the double‐headed arrow in figure 1. Author should describe more details explain of inflammation mediated COPD mechanism.

We added a detailed description of Figure 1 to the “Introduction” section as proposed. Please also see our answer to the first point of Reviewer 1.

3. Lane 132‐133 This table 1 is very indistinct. Please include the reference number in the table.

We thoroughly revised Table 1 and added references to every comment as proposed. Moreover, a statement concerning the long‐term use of ICS was added (please also see our answer to Reviewer 3, point 1).

4.Lane 237‐248 This paper lacks in‐depth summary. Please discuss that why cardiovascular diseases are associated closely to COPD and inflammation. I recommend to revise the manuscript so that the purpose appears more clearly.

We agree that our summary lacks an in‐depth discussion and therefore revised the respective section. Statements concerning the link between COPD, cardiovascular disease and systemic inflammation were added as requested.

Reviewer 3 Report

This is a well written paper dealing with cardiovascular comorbidity and COPD. I have only one criticism about the following statement: "This finds its expression in elevated markers of inflammation even in pulmonary stable disease [7] indicating a link between local  pulmonary and systemic inflammatory processes. Hence, therapeutic interventions based on inhaled corticosteroids (ICS) stands to reason as they are a cornerstone of anti-inflammatory therapy in lung disease in general".Comment:this statement is completely wrong: ICS are the cornerstone in asthma treatment while their use in COPD is deserved to stage D of the disease classified according to GOLD 2018 or in group C when exacerbations persist despite the inhaled therapy with LABA+LAMA. The same misleading concept is reported in table 1 first line. Furthermore conflicting results are reported concerning ICS and pneumonia in COPD patients.The generalization of ICS role in lung inflammation is trivial. Distinct patways of inflammation have been described in asthma and in COPD, the latter not suscettible to respond to ICS therapy unless in some defined "endtoypes-phenotypes" of the disease (ie chronic bronchitis with eosinophylic infammation).

This is a well written paper dealing with cardiovascular comorbidity and COPD. I have only one criticism about the following statement: "This finds its expression in elevated markers of inflammation even in pulmonary stable disease [7] indicating a link between local pulmonary and systemic inflammatory processes. Hence, therapeutic interventions based on inhaled corticosteroids (ICS) stands to reason as they are a cornerstone of anti-inflammatory therapy in lung disease in general". Comment: this statement is completely wrong: ICS are the cornerstone in asthma treatment while their use in COPD is deserved to stage D of the disease classified according to GOLD 2018 or in group C when exacerbations persist despite the inhaled therapy with LABA+LAMA. The same misleading concept is reported in table 1 first line. Furthermore, conflicting results are reported concerning ICS and pneumonia in COPD patients.

We agree to this important point and adapted the paragraph accordingly. Moreover, we summarized the most recently published 2019 GOLD report highlighting the importance of exacerbations or blood eosinophilia for prescribing ICS. Furthermore, Table 1 was revised accordingly and a statement concerning the long‐term use of ICS was added to the description with reference to the text (please also see Reviewer 2, point 3).

The generalization of ICS role in lung inflammation is trivial. Distinct pathways of inflammation have been described in asthma and in COPD, the latter not susceptible to respond to ICS therapy unless in some defined "endotypes‐phenotypes" of the disease (ie chronic bronchitis with eosinophilic inflammation).

We agree that our description of using ICS in lung inflammation was trivial. Therefore, we revised the respective parts of section “Cardiovascular risk” accordingly. This does not only include the updated GOLD 2019 report, but also a recent study investigating the use of FeNO levels to predict treatment response to ICS in patients with non‐specific respiratory symptoms. The latter underlines the role of ICS in targeting local eosinophilic airway inflammation and therefore the need to better endotype/phenotype COPD as stated in your valuable comment.